# SWI/SNF Complex Alterations in Tumors with Rhabdoid Features: Novel Therapeutic Approaches and Opportunities for Adoptive Cell Therapy

**DOI:** 10.3390/ijms241311143

**Published:** 2023-07-06

**Authors:** Juan José Soto-Castillo, Lucía Llavata-Marti, Roser Fort-Culillas, Pablo Andreu-Cobo, Rafael Moreno, Carles Codony, Xavier García del Muro, Ramon Alemany, Josep M. Piulats, Juan Martin-Liberal

**Affiliations:** 1Medical Oncology Department, Catalan Institute of Oncology (ICO), 08908 Hospitalet de Llobregat, Spain; garciadelmuro@iconcologia.net (X.G.d.M.); jmpiulats@iconcologia.net (J.M.P.); 2Medical Oncology Department, Catalan Institute of Oncology (ICO), 17007 Girona, Spain; lucia.llavata@hotmail.com (L.L.-M.); rfort@iconcologia.net (R.F.-C.); 3Medical Oncology Department, Parc Tauli Hospital Universitari, 08208 Sabadell, Spain; pabloandreu94@gmail.com; 4Cancer Immunotherapy Group, iPROCURE Program, Bellvitge Biomedical Research Institute (IDIBELL), Catalan Institute of Oncology (ICO), 08908 Hospitalet de Llobregat, Spain; rafamoreno@iconcologia.net (R.M.); ccodony@idibell.cat (C.C.); ralemany@iconcologia.net (R.A.)

**Keywords:** SWI/SNF complex, *SMARCA4*, *SMARCB1*, *ARID1A*, rhabdoid tumors, epigenetic drugs, immunotherapy, adoptive cell therapy

## Abstract

The SWItch/Sucrose Non-Fermentable (SWI/SNF) chromatin-remodeling complex is one of the most remarkably altered epigenetic regulators in cancer. Pathogenic mutations in genes encoding SWI/SNF-related proteins have been recently described in many solid tumors, including rare and aggressive malignancies with rhabdoid features with no standard therapies in advanced or metastatic settings. In recent years, clinical trials with targeted drugs aimed at restoring its function have shown discouraging results. However, preclinical data have found an association between these epigenetic alterations and response to immune therapy. Thus, the rationale for immunotherapy strategies in SWI/SNF complex alteration-related tumors is strong. Here, we review the SWI/SNF complex and how its dysfunction drives the oncogenesis of rhabdoid tumors and the proposed strategies to revert this alteration and promising novel therapeutic approaches, including immune checkpoint inhibition and adoptive cell therapy.

## 1. Introduction

Epigenetic reprogramming mechanisms have recently been identified as a new hallmark of cancer [1]. These mechanisms go beyond pathogenic mutations in oncogenes and tumor suppressor genes mutations since the packaging and assembly of the DNA molecule and its interactions with histones are important steps in the regulation of gene expression.

The SWItch/Sucrose Non-Fermentable (SWI/SNF) complex (SWI/SNFc) is a family of ATP-dependent chromatin remodeling complexes found in eukaryotes. Its main function is to regulate histone–DNA interactions in the reassembly of nucleosomes using the energy released by ATP hydrolysis. The dynamics among nucleosomes act as ejection and/or sliding motions that translate into easier or harder access to chromatin, allowing genes to be activated or repressed [2].

Data from The Cancer Genome Atlas (TCGA) project have shown that mutations in genes encoding subunits of SWI/SNFc are present in nearly 25% of all cancers [3,4]. In vitro and in vivo studies support that SWI/SNF mutations are tumor-promoting, as the majority of these alterations produce a loss-of-function phenotype. Consequently, most of the genes involved in this complex are considered to be tumor suppressors [5,6].

SWI/SNFc alterations were first implicated in oncogenesis after the discovery of *SMARCB1*. Biallelic-inactivating mutations in *SMARCB1* were characterized in 1998 in malignant rhabdoid tumors, an aggressive type of pediatric soft-tissue sarcoma [7]. It has also been shown that genetically engineered mice with inactivation of this gene rapidly develop cancer with 100% penetrance [8]. With the advent of data from TCGA, other genes were added to the list of solid tumors harboring mutations in genes encoding the SWI/SNFc [9,10,11,12]. Some of these tumors are considered extremely rare and have a poor prognosis. Moreover, the few therapeutic options—mainly chemotherapy—for the treatment of advanced/metastatic disease have shown poor results, so enrollment of these patients in clinical trials is encouraged. Currently, drugs targeting epigenetic and DNA repair pathways are being tested, both in monotherapy and in combination with tyrosine kinase inhibitors (TKI) or chemotherapy. However, clinical results have not been as good as expected.

One of the most exciting vulnerabilities that have recently emerged is the link between SWI/SNF aberrations and immunogenicity. For example, *SMARCB1*-mutant rhabdoid tumors are infiltrated by clonally expanded populations of T lymphocytes, suggesting a tumor-specific immune response [13,14]. These and several other studies broaden the horizons for further research into how SWI/SNFc-mutant monogenic tumors might be sensitive to novel therapeutic strategies, such as immune-checkpoint inhibitors (ICI) or adoptive cell therapy (ACT). In this review, we discuss the role of SWI/SNc alterations in rare rhabdoid tumors and provide the latest evidence on new therapeutic approaches.

## 2. The SWI/SNF Complex

### 2.1. Description

DNA as a naked chain of nucleotides is extremely unusual in cells. For most of the cell cycle, this molecule is bound to histones and other proteins so that a few meters of nucleic acid are stored in a compact shape inside the nucleus as chromatin.

There are many protein complexes that play a key role in chromatin remodeling. The SWI/SNFc is one of them. It consists of an evolutionarily conserved family of enzymes broadly found in eukaryotes, which essentially work in an ATP-dependent manner, using the energy from its hydrolysis to modify the interactions between histone and nucleosomes, as well as the distance between them, thereby changing the way DNA is packaged [15].

The SWI/SNFc was first identified in yeast. This complex has been conserved across lineages and species as an essential factor in controlling chromatin accessibility. However, it has diverged into different subclasses with species-specific subunits, resulting in a heterogeneous array of regulatory proteins [16]. SWI/SNFc integrates paralogous subunits (homologous genes with different origins within the same genome) according to specific cell types or during certain developmental processes, such as those mediating cell differentiation (e.g., in hematopoietic cells, osteoblasts, skeletal muscle, among others), or lineage specification in embryonic stem cells [16,17,18]. Thus, the enormous diversity of SWI/SNF subunits composition leads to many functional implications in the transcriptional regulation of genes. The location of regulators provides diversity in how chromatin can be transcribed. Therefore, the interaction of SWI/SNF with enhancers or promoters of genes can have multiple transcriptional consequences [19,20].

### 2.2. Proteins Involved and Mechanism of Action

Based on their composition, SWI/SNF complexes in mammals are divided into three major subfamilies: BR-associated factor (BAF, or SWI/SNF-α), polybromo BRG1-associated factor (PBAF, or SWI/SNF-β), and non-canonical BAF (ncBAF/GBAF) [21]. All complexes contain three types of subunits: the ATPase subunits (*SMARCA4*/BRG1 or *SMARCA2*/BRM), which are present in a mutually exclusive manner and harbor the catalytic function, the core subunits (*SMARCC1*/BAF155, *SMARCC2*/BAF170, and *SMARCB1*/BAF47/INI1), which are important for assembly and stabilization of the ATPase, and the complex-specific subunits (e.g., *ARID1A*/BAF250A and *ARID1B*/BAF250B for BAF, *ARID2*/BAF200 and *PBRM1*/BAF180 for PBAF, or *BICRA*/GLTSCR1 for ncBAF, among others). Many other proteins are shared among BAF, PBAF, and ncBAF, so the SWI/SNFc may be variable in composition and be encoded by multiple genes (Figure 1). The characteristic combination of these proteins is essential for the identity of these three complexes and their diverse biological functions [16,19,22]. Figure 2 represents general actions and consequences of proficient and deficient SWI/SNF.

Despite this variable composition, the structure of the complexes is largely conserved. Structural models in animals have shown that the SWI/SNFc envelops the nucleosome by forming a clamp shape, with the ATPase and core subunits in contact with the nucleosome. However, SWI/SNFc has no intrinsic ability to bind DNA, but it tends to be recruited by transcription factors close to gene promoters. The core subunits act as a hinge to stabilize the connection between the nucleosome and the complex, while the ATPase can recognize the superhelicoidal location 2.5 (SHL2.5) domain within the DNA, where the energy released from ATP hydrolysis breaks the contacts between histones and DNA [23,24]. This causes localized disruption to the chromatin structure, making it more accessible to transcriptional regulators and RNA. Chromatin remodeling is carried out by dynamic forces that slide, displace, or destabilize nucleosome components or even eject histone dimers, leading to the regulation of transcription in large sets of genes [25,26]. cBAF is mainly active at enhancers, whereas PBAF and ncBAF are reported to be enriched at promoters, although they can also bind to some enhancers [27].

Bromodomains in *SMARCA4* and *SMARCA2* play an important role. They typically bind to acetylated lysines on the N-terminal tails of histones H3 and H4, as well as other proteins. This binding is critical for stable interaction with promoters, which are essential for differentiation-specific gene programs [28]. Bromodomain-containing proteins have been classified into different groups based on their structure. *SMARCA2*, *SMARCA4*, and *PBRM1* belong to family VIII of bromodomains, which are located in the C-terminal region of the protein [29].

### 2.3. Role in Carcinogenesis

SWI/SNF plays a critical role in various cellular processes, such as cell cycle control, cell differentiation, apoptosis, or metabolism.

In 1997, Trouche et al. showed that retinoblastoma (RB) protein must cooperate with BRG1 to induce complete G1 arrest through inhibition of the E2F1 transcription factor [30]. A few decades later, Ruijtenberg et al. demonstrated that SWI/SNFc could induce cell cycle arrest in proliferating muscle cell precursors. Loss of SWI/SNF function also impaired the specific gene expression in differentiated cell types, also promoting failure to exist in the cell cycle [31]. Further research has evidenced the key role of some SWI/SNF subunits in cell cycle regulation by promoting arrest [32,33] or programmed cell death [34].

It has been proposed that SWI/SNFc is also involved in many key differentiation and developmental processes in mammalian tissues, such as adipocytes, hematopoietic cells, neurons, osteoblasts, or muscle cells [24]. Klochendler–Yeivin et al. showed that loss-of-function mutations in the SNF5 gene were detrimental to the early development of cell embryos, with lethality in nullizygous blastocysts [6].

DNA damage repair (DDR) is another well-established non-transcriptional function involving SWI/SNF. Between 2009 and 2010, a number of authors reported that BAF and PBAF could be gathered around sites of DNA damage, including the phosphorylation of BAF170 dependent on ATM and ATR, or the existence of cooperative structures between SWI/SNF and γ-H2A.X for the repair of DNA double-strand breaks (DSBs) [35,36]. DSBs are one of the most deleterious forms of DNA damage, leading to genome instability if not repaired.

There is a robust signaling cascade initiated in response to DNA DSBs that can lead to transcriptional upregulation of repair genes, cell cycle arrest, and, in some cases, programmed cell death, in which the SWI/SNFc can participate [37]. Once DNA DSBs are detected and signaled, two major repair pathways are activated: non-homologous end joining (NHEJ) and homologous recombination (HR). SWI/SNFc has been implicated in both, as well as in other cellular pathways, such as alternative end joining, although its role in these is limited. Current evidence suggests that BAF plays a critical role in the process of NHEJ. Specifically, BAF is required to restructure chromatin adjacent to DNA DSBs in order to facilitate the binding of repair factors [38].

The role of chromatin remodeling complexes in the process of HR—particularly the SWI/SNFc—is important, given the requirement for manipulation of the chromatin flanking the DSB and the sister chromatid during strand invasion [35]. In addition, the HR process does not function effectively in cells lacking SWI/SNFc subunits [39,40]. Therefore, the loss of SWI/SNFc function can lead to defective DNA repair and increased sensitivity to DNA damage.

SWI/SNF also has a role in maintaining chromosomal stability. First, many specific regions critical for chromosome organization, such as the binding sites of CTCF and cohesins, are enriched with BAF and PBAF, suggesting that these complexes are critical for regulating the overall chromatin structure [41,42]. Moreover, Brownlee et al. found that a deleterious function of *PBRM1* facilitated aneuploidy due to its role in sister chromatid cohesion [43].

Crosstalk with key proliferation, survival, and cell cycle control pathways has also been described. Recent evidence has shown that SWI/SNFc may interact with canonical proto-oncogenes such as *MYC* and tumor suppressor genes such as *TP53*, *CDKN2A*, *RB1*, or *BRCA1* [44,45,46,47]. Although the nature and impact of these interactions on carcinogenesis remain poorly understood, it is well established that the SWI/SNFc plays a crucial role in facilitating p53’s ability to mediate gene expression and exert its tumor suppressor functions. In addition, this interaction is relevant for the regulation of VEGFR2 through chromatin remodeling [48].

Figure 3 summarizes cellular processes in which SWI/SNF is involved.

Given that large-scale cancer analyses, such as the TCGA, have demonstrated a high rate of mutations in genes involved in SWI/SNFc (nearly 25% in more than 100,000 tumors), there is a strong rationale for accelerating research and finding therapeutic approaches that target SWI/SNF aberrations. Loss-of-function mutations are the most common alterations leading to SWI/SNF inactivation, but the TCGA also revealed other genomic aberrations, like amplifications or overexpression. Consequently, mutations in SWI/SNF genes can lead to tumor suppressor or oncogene functions, the former being the most commonly described in the literature [49]. Biallelic inactivation of *SMARCB1* due to nonsense mutations or gene deletions, resulting in complete loss of protein, was the first SWI/SNF alteration found [50]. In contrast to this tumor suppressor role, gain-of-function of *ARID1A* and *SMARCA4* was found in hepatocellular carcinoma and in breast cancer, respectively, suggesting that SWI/SNFc alterations could also act as oncogenes [51]. Deregulation of *SMARCA4* has also been described in lung adenocarcinoma, medulloblastoma, pancreatic adenocarcinoma, and Burkitt’s lymphoma, with a variable mutation rate [52,53,54,55]. *ARID1A* mutations have also been reported in gastric cancer, colorectal cancer, and cholangiocarcinoma [56,57,58].

In summary, cellular function and cell cycle phase appear to be important in defining the potential consequences of SWI/SNFc alterations in human tumors.

## 3. Rhabdoid Tumors Associated with SWI/SNF Complex Alterations

Mutations in SWI/SNF can arise either during tumor development or as an initiating oncogenic driver event. Solid tumors with rhabdoid features are one of the best examples of SWI/SNF disruption leading to carcinogenesis, sometimes by well-characterized complete loss of *SMARCA4* or *SMARCB1*. These SWI/SNF-mutated monogenic diseases correspond to a large number of solid malignancies with an aggressive clinical course and an ominous prognosis but also with a clear target to be reverted.

Etymologically, “rhabdoid” comes from the Greek term rhabdoeidēs, meaning “rod-shaped”. In Pathology, this feature is characterized by the presence of sheets and clusters of variably cohesive, large cells (sometimes with prominent nucleoli) and large, paranuclear intracytoplasmic hyaline globules [59]. Rhabdoid tumors include a few rare and aggressive tumors, most of which are sarcomas, affecting mainly the pediatric population.

Epithelioid sarcoma (ES): ES is a rare and aggressive soft tissue sarcoma (<1% of all sarcomas) of young adulthood. Characteristically, it grows as a painless mass in the distal limbs. Up to 30–50% of cases may present with lymph node or visceral metastases. This tumor is molecularly characterized by a complete lack of *SMARCB1* expression, identified as loss of INI1 immunohistochemical stain in 90% of cases [60,61]. Biallelic-inactivating *SMARCB1* mutations are the most frequent aberration. Nonsense frameshift and splice site mutations complete the causes of *SMARCB1* loss, but these are quite rare [62,63,64]. Preclinical data have suggested that some miRNAs (miR-193a-5p, miR-206, miR-381, miR-671-5p) are involved in *SMARCB1* inactivation through epigenetic mechanisms [65,66,67]. However, this hypothesis remains to be confirmed. *SMARCB1* deficiency disrupts essential molecular pathways of cell cycle control, gene transcription, and cell survival, leading to upregulation of *MYC*, Wnt/β-catenin, and Sonic Hedgehog signaling [68,69,70] and enhancing the oncogenic process. In particular, the antagonistic activity between *SMARCB1* and EZH2, the catalytic subunit of Polycomb Repressive Complex 2 (PRC2), has provided the rationale for the approval of tazemetostat in these patients, as discussed later.

Malignant rhabdoid tumor (MRT): This is a rare and highly malignant neoplasm of childhood (usually occurring within the first 2 years of life). Regardless of location, all rhabdoid tumors are highly aggressive and have a poor prognosis. Anatomically, MRT can arise from any site, but the central nervous system (CNS) is the most common site, followed by the kidney and soft tissues. Nearly 95% of MRTs have mutations in *SMARCB1*, and in very few cases, *SMARCA4* is altered (<5%) [71]. Complete inactivation of *SMARCB1* has been associated with large deletions of chromosome 22, whole exon duplications or deletions, and point mutations leading to stop codons [72]. A predisposition syndrome has also been described in families with germline mutations in these genes [73], requiring a second somatic hit given the tumor suppressor nature of *SMARCB1*. Loss of *SMARC* leads to destabilization of SWI/SNF function and results in a more hypomethylated state of the chromatin in cell lines [74]. Despite these homogeneous molecular features, MRT phenotypes are highly variable, and many subclassifications have been proposed [14]. Multimodal therapy, including radical surgery followed by chemotherapy, intrathecal methotrexate (in CNS MRT), and radiotherapy, is the recommended approach. Recently, tazemetostat has shown good results on MRT [63].

Small-cell carcinoma of the ovary, hypercalcemic variant (SCCOHT): SCCOHT is a very rare tumor of the ovary (<0.5% of malignant ovarian tumors), mostly affecting women under the age of 30. A painful pelvic mass associated with hypercalcemia should prompt consideration of this diagnosis, as this endocrine disorder is present in 60% of patients [75,76]. The prognosis is poor, with a long-term survival rate of approximately 30%. More than 95% of SCCOHTs harbor deleterious mutations in *SMARCA4* (biallelic in 25%), and these have been described in both germline and somatic lines [77]. Loss of *SMARCA2* by epigenetic inactivation [78] and *SMARCB1*/*ARID1A* inactivating mutations may also occur but are extremely rare [75,79]. Cytotoxic chemotherapy is the cornerstone of treatment in the advanced setting, but further research is needed due to poor outcomes. In an early clinical trial, tazemetostat was shown to control disease in some anecdotal cases [63]. Different approaches targeting histone deacetylases, tyrosine kinase receptors (TKR), and ICI are under investigation.

Renal medullary carcinoma (RMC): RMC is a rare tumor that originates in the kidney and tends to be aggressive and resistant to standard therapy. It usually occurs in young patients who are characteristically affected by sickle cell traits or other hemoglobinopathies [80]. Molecularly, RMC shows a complete loss of *SMARCB1* expression, which may occur due to inactivating translocations or deletions [81]. The hypertonic environment of the interstitial space within the renal medulla is thought to be a major trigger for DNA double-strand breaks, which, in turn, would be the underlying mechanism of *SMARCB1* inactivation [67].

Malignant peripheral nerve sheath tumor (MPNST): This is an invasive soft tissue sarcoma arising from both malignant schwannoma cells and malignant rhabdomyoblasts. Approximately 50% of cases are associated with neurofibromatosis type 1 [82]. The combined loss of *NF1* and *SMARCB1* is explained by large deletions or biallelic mutations following a sequential “4-hit” mechanism. In addition, some studies have reported that germline mutations in *SMARCB1* can cause familial schwannomatosis and meningiomas [83]. MPNST also frequently shows chromosomal abnormalities, such as copy number variants (CNVs) of chromosome 17 or 9p21 losses and aberrations in *TP53*, *CDKN2A*, *SUZ12*, and *RASSF1* (the latter two are core components of the PRC2). Several single nucleotide variants (SNVs) have been found in TKR, but their oncogenic contribution remains unclear [84].

Myoepithelial carcinoma: This is a rare tumor that mainly affects the salivary glands, breast, soft tissues, and, less commonly, other organs such as the lungs. It may mimic a pleomorphic adenoma but has rhabdoid features and *SMARCB1* mutations [85].

Extra-skeletal myxoid chondrosarcoma (ESMC): This is a sarcoma with a propensity for local recurrence and development of metastases despite an indolent clinical course. It is resistant to chemotherapy. This tumor is associated with tumor-specific translocations involving the Ewing’s Sarcoma (EWS) gene and not infrequently with *SMARCB1* mutations [86].

Poorly differentiated chordoma: This tumor is rare, usually affects adults, and arises at the base of the skull and spine. Cases of chordoma diagnosed at a young age are often poorly differentiated, with cytologic atypia, increased cellularity, and mitosis, and their aggressive behavior is associated with a high incidence of metastatic and short patient survival. Recent studies have described the loss of *SMARCB1* in poorly differentiated chordomas [87].

Rhabdomyosarcoma: This is the most common sarcoma in childhood, accounting for up to 5–10% of all pediatric malignancies. Four subtypes have been described, all with different clinical features. Molecularly, rhabdomyosarcoma can be characterized by oncogenic drivers, such as *FOXO1* fusions, *MYOD1* mutations, *VGLL2* fusions, and *TFCP2* fusions. The embryonal subtype harbors mutations affecting the RAS family proteins and *TP53* [88]. No specific SWI/SNFc aberrations characterize this tumor, but recent data support the importance of *SMARCA4* expression in the maintenance of alveolar and embryonal rhabdomyosarcoma cells. In these cases, *SMARCA4* expression favors cell growth, as knockdown of the gene experimentally affected the viability of cell lines. On the other hand, high expression of *SMARCA2* has been associated with reduced survival in a cohort of patients [89].

## 4. Therapeutic Approaches

### 4.1. Strategies Focusing on SWI/SNFc and Related Targets

Table 1 lists early clinical trials testing drugs in SWI/SNF-altered solid tumors, with a focus on rhabdoid tumors. Different therapeutic strategies have been grouped into five categories according to their mechanism of action or targeted pathway, as described below.

#### 4.1.1. Targeting SWI/SNF Subunits

Mutations in genes encoding specific SWI/SNF subunits create shared dependencies with other subunits and partners of the complex, conferring a vulnerability that can promote synthetic lethal mechanisms. For example, a mutated subunit would not completely disable SWI/SNF function because it could be partially compensated by its paralogue gene. However, if both deficiencies occur together, this would lead to cell death, causing lethality [90]. In rhabdomyosarcoma, dual depletion of *SMARCA4* and *SMARCA2* by protein degradation enzymes has been shown to inhibit tumor growth. Also, *SMARCA4* inhibition with histone deacetylase 3 is another approach to target SWI/SNF in wild-type tumors.

Several intra- and inter-complex vulnerabilities have been found to be associated with synthetic lethal phenotype in SWI/SNF-deficient tumors. Examples of intra-complex dependencies are mutations of *ARID1A* with its paralogue *ARID1B* or mutations of *SMARCA4* with *SMARCA2*. There are also extra-complex dependencies, such as mutations of *SMARCB1* with its no-paralogue gene *BRD9* [27].

Drugs targeting SWI/SNF have mainly focused on the ATPase and *SMARCA2/4* bromodomain subunits, seeking synthetical lethality by inactivating specific proteins of the complex or neutralizing their overexpression. Furthermore, there are some ongoing trials testing *BRD9* inhibitors in tumors with *SMARCB1* loss or *SMARCA2* degradation in *SMARCA4*-mutant tumors (Table 1).

#### 4.1.2. Targeting PRC via EZH2

SWI/SNF and PRC have opposing gene-regulatory functions. While SWI/SNF normally locates at sites marked by histone H3K27 and cooperates with transcription factors to open chromatin for transcription, PRC (mainly PCR2) acts through its enzymatic subunit EZH2, blocking H3K27 by methylation and repressing transcription.

There is an antagonistic relationship between EZH2 and *SMARCB1* (as well as other subunits of the SWI/SNFc), resulting in genetic dependence on EZH2 in some SWI/SNF-mutant cancers. *SMARCB1* inactivation results in increased levels of methylated H3K27 (promoted by PCR2) [27]. High levels of EZH2 often correlate with advanced tumor stage and poor prognosis, so inhibition of EZH2 may block proliferation and survival. Currently, EZH2 is a therapeutic target with an approved drug, tazemetostat. It was approved by the Food and Drug Administration (FDA) in January 2020 following the results of a phase 2 clinical trial in advanced ES with loss of INI1/*SMARCB1*. Tazemetostat activity was evaluated in 62 patients with metastatic or locally advanced ES with INI1 loss. The overall response rate (ORR) was 15%. Of these responses, 67% lasted ≥ 6 months with acceptable tolerability [63].

Given this benefit of tazemetostat, many other ongoing phase 1 and 2 trials are assessing its activity in rhabdoid tumors (Table 1).

#### 4.1.3. Targeting DDR Process

Non-transcriptional roles of SWI/SNF in DNA repair may be exploited therapeutically, as loss-of-function mutations in SWI/SNF genes could be potential biomarkers for inhibitors of the DNA damage response, such as PARP inhibitors (Table 1).

#### 4.1.4. Targeting TKR

The presence of links to other known oncogenic pathways (*MYC*, *RAS*) may serve as therapeutic targets that could potentially be inhibited by small molecules. However, most SWI/SNF proteins act as tumor suppressors, making restoration of their function with current drugs extremely challenging.

Interestingly, a dependency between rhabdoid tumors and *ARID1A*-mutated ovarian cancer and some TKRs has been described, especially EGFR and HER2, but also FGFR, IGF, and c-MET [27,91]. As a result, studies are underway to evaluate the blockade of aberrant TKR in these tumors (Table 1).

#### 4.1.5. Targeting Kinases Involved in Cell Cycle

CDK4/6 inhibitors and other agents that may induce synthetic lethality by impairing DDR (ATR inhibitors and platinum chemotherapy) are also under investigation. A synthetic lethal interaction between *SMARCA4* loss and CDK4/6 inhibition, mediated by cyclin D1 deficiency, has been demonstrated. Loss of *SMARCA4* causes downregulation of cyclin D1, and there are data supporting that this vulnerability mediates drug sensitivity to CDK4/6 inhibition in *SMARCA4*-deficient cancer cells [92]. Geoerger et al. evaluated the safety and showed preliminary activity of ribociclib in pretreated neuroblastomas, MRT, and other cancers with documented cyclin D–CDK4/6–INK4–Rb pathways aberrations, including those with *SMARCB1* loss [93].

**Table 1 ijms-24-11143-t001:** Clinical trials targeting SWI/SNF subunits and other molecular pathways in solid tumors harboring SWI/SNFc alterations.

Author/Year	NCT	Study Design	N	Tumor	Drug	Endpoints/Results and Grade 3–5 AEs

Targeting SWI/SNF complexes
Ongoing	05639751	Phase I	86	Advanced *SMARCA4*-mutant solid tumors	PRT3789 (*SMARCA2* degrader)	Safety (DLT, MTD, AEs). PK, PD. Efficacy (ORR, PFS, DOR, BOR).
Ongoing	04965753	Phase I	104	Advanced synovial sarcoma and advanced *SMARCB1*-loss tumors	FHD-609 (*BRD9* inhibitor)	Safety (TRAEs, AEs, DLTs). PK, PD. Efficacy (ORR, DOR, PFS, OS).
Ongoing	05355753	Phase I/II	110	Adolescents and adults with advanced *SMARCB1*-altered tumors	CFT8634 (*BRD9* inhibitor)	Safety (AEs, DLTs). PK, PD. Efficacy (ORR, DOR, PFS, OS).
Ongoing	03297424	Phase I/II	60–136	Advanced malignancies with a known *ARID1A* mutation	PLX2853 (BET inhibitor)	Safety (DLT, AEs). PK, PD. Efficacy (ORR, DOR, OS, PFS).

Targeting PCR via EZH2
Gounder et al. (2020) [63]	02601950	Phase II	62	Advanced epithelioid sarcoma with loss of INI1/*SMARCB1*	Tazemetostat	Efficacy: ORR 15%; DOR not reached; median PFS 5.5 months, median OS 19 months. AEs: Anemia G3 (6%), weight loss G3 (3%). No G4-5 AEs.
Ongoing	01897571	Phase II	420	Advanced-stage solid tumors or B cell lymphomas	Tazemetostat	Efficacy. Safety (MTD, bioavailability).
Ongoing	02601937	Phase I	82	Children with MRT, ATR, RTK, and other tumors with rhabdoid features	Tazemetostat	Safety (AEs)
Ongoing	03213665	Phase II	49	Children R/R solid tumors, NHL or histiocytic disorders EZH2, *SMARCB1,* or *SMARCA4*-mutated	Tazemetostat	Efficacy (ORR, PFS). Safety (AEs).
Ongoing	02601950	Phase II	250	Adults MRT, ATRT, RTK with loss of *SMARCB1* or *SMARCA4* or EZH2-mutated tumors	Tazemetostat	Efficacy (ORR, DOR, PFS). Effect of tazemetostat on immune priming.
Ongoing	02875548	Phase II	300	Adults MRT, ATRT, RTK, synovial, or epithelioid sarcoma, mesothelioma, DLBLC	Tazemetostat	Efficacy (PFS, OS). Safety (AEs).
Ongoing	05151588	Phase II	30	Locally Advanced *SMARCB1*-deficient sinonasal carcinoma	Chemotherapy + Tazemetostat	Efficacy (BOR, PFS, OS, orbit preservation rate). Safety (AEs).
Ongoing	02601937	Phase I	109	Children and adolescents with R/R INI1-negative tumors or synovial sarcoma	Tazemetostat	Safety (MTD, AEs). PK. Efficacy (ORR, PFS, OS).

Inhibitors of DNA damage repair
Ongoing	0405269	Phase II	40–116	*ARID1A*-deficient gynecological tumors	Celasertib ± Olaparib	Efficacy (BOR)
Ongoing	03682289	Phase II	89	BAF250-negative solid tumors	Celasertib ± Olaparib or Durvalumab	Efficacy (ORR, DOR, PFS, OS) Safety (AEs)
Ongoing	03207347	Phase II	57	Adults with BAP1 and *ARID1*-mutant tumors	Niraparib	Efficacy (ORR, PFS, OS)
Ongoing	02576444	Phase II	64	Adults with cancer containing mutations in homologous DNA repair or other DDR genes, including *ARID1A*	Olaparib + Capivasertib	Efficacy (ORR)
Ongoing	04065269	Phase II	40	Adults with relapsed gynecological cancers, with or without loss of *ARID1A*	Olaparib + Ceralasertib (AZD6738)	Efficacy (ORR, DCR, PFS, TTP, OS)
Ongoing	05523440	Phase II	92	Recurrent ovarian or endometrial cancer with *ARID1A* mutation	Bevacizumab ± Niraparib	Efficacy (ORR, DOR, PFS). Safety (AEs)

Targeting tyrosine kinase receptors
Ongoing	03718091	Phase II	223	Adults with advanced-stage solid tumors (including an *ARID1*-mutant cohort)	Berzosertib (VX-970 and M6620; ATR)	Efficacy (changes in Phospo-CHK1, ɣH2AX levels and DCR
Ongoing	02059265	Phase II	35	Adults with recurrent or persistent gynecological cancer, with or without BAF250 loss	Dasatinib	Efficacy (ORR, PFS, OS) Safety
Ongoing	04284202	Phase II	30	Adults with NSCLC *ARID1*-mutant	Dasatinib + Toripalimab	Efficacy (PFS, OS)

Targeting kinases involved in the cell cycle
Ongoing	02114229	Phase II	180	Children and young adults with ATR and or extra-CNS MRT (with loss of *SMARCB1* and/or extra-CNS MRT (with loss of *SMARCB1* or *SMARCA4*)	Alisertib (Aurora A inhibitor)	Efficacy (ORR, PFS); PK; PD
Geoerger et al. (2017) [93]	01747876	Phase I	32	Children and young adults with *SMARCB1*-loss tumors	Ribociclib	Efficacy: ORR 0% AEs: Neutropenia G3-4 (63%), leukopenia G3-4 (38%), thrombopenia G3-4 (28%), fatigue G3-4 (3%), AST increased G3-4 (3%), anemia G3-4 (3%), decreased appetite G3-4 (3%)

Abbreviations: AEs: adverse events; ATRT: atypical teratoid rhabdoid tumor; BOR: best overall response; DCR: disease control rate; DLBLC: diffuse large B cell lymphoma; DLT: dose-limiting toxicity; DOR: duration of response; MTD: maximum tolerated dose; NHL: non-Hodgkin lymphoma; ORR: overall response rate; OS: overall survival; PD: pharmacodynamics; PFS: progression-free survival; PK: pharmacokinetics; R/R: recurrent or relapsed; RTK: rhabdoid tumor of the kidney; TRAEs: treatment-related adverse events.

### 4.2. Immunotherapy Strategies

The first immunotherapy was approved for the treatment of an advanced solid tumor more than three decades ago [94]. Since then, cancer immunotherapy has become one of the cornerstones of cancer treatment, particularly with the advent of ICI. Today, the number of approvals in both advanced and adjuvant settings is increasing as a number of antibodies targeting the programmed cell death protein 1 (anti-PD-1) or its ligand (anti-PD-L1) are being tested in several phase III trials [95,96].

SWI/SNFc aberrations were first proposed as potential predictors of response in a retrospective analysis of patients with advanced renal cell carcinoma treated with ICI. *PBRM1* truncating or loss-of-function mutations were demonstrated to be involved in the activation of several pathways (IL6/JAK-STAT3, TNFα signaling via NF-κB and hypoxia-responsive signature), especially in the context of immunostimulatory responses, and thus associated with responsiveness to nivolumab (anti-PD1). Clonality and a high proportion of cells harboring this *PBRM1*-deficient alteration were associated with tumor responses [97].

In addition, Pan et al. showed how the inactivation of *ARID2* or *BRD7* in melanoma cell lines attracted effector T cells via interferon (IFN)-γ signaling [97]. Other preclinical studies have associated *ARID1A*-mutant cancers with an increased number of TIL, higher PD-L1 expression, and the benefit from ICI treatment [98,99,100].

Rhabdoid tumors with specific SWI/SNFc subunit mutations (mostly *SMARCB1* and *SMARCA4*) have been shown to be associated with immune infiltration in the tumor microenvironment, monocytes/macrophages and CD8+ T lymphocytes being the two most predominant subtypes. *SMARCB1* deficiency seems to impact critically on epigenetic regulation and immunogenicity, as *SMARCB1* mutation is the predominant genomic aberration able to trigger the IFN pathway in these tumors. These preliminary results require prospective clinical validation [13].

Those findings may be applicable to other rare malignancies. For example, Leruste et al. reported complete responses in mice with rhabdoid tumors receiving anti-PD1, and Jelinic et al. published durable responses in four patients with SCCOHT treated with anti-PD1 monotherapy, most of whom had deleterious mutations of *SMARCA4* [13,101]. An RNA sequencing study in mouse cell lines found a correlation between lower expression of *ARID2* and *PBRM1* and higher CD8+ T cell-mediated cytotoxicity. Furthermore, data extracted from TCGA melanoma patients showed that *ARID2* mRNA levels correlated with survival depending on CD8+ expression [98]. The same investigators demonstrated the enrichment of dendritic cells and a favorable ratio of M1-like macrophages to tumor-promoting M2-like macrophages in *PBRM1*-deficient tumors, providing additional evidence for the use of immunotherapy targeting the tumor microenvironment.

Therefore, previously described alterations in individual SWI/SNF proteins may confer susceptibility to immunotherapy in rhabdoid tumors with scarce therapeutic options and poor prognosis.

#### 4.2.1. Immune-Checkpoint Inhibition

Preclinical evidence and case reports of susceptibility to ICI are increasingly being reported. Recently, Wang et al. published the first study of alterations in the 31 members of the SWI/SNFc and their association with ICI outcomes, demonstrating favorable overall survival (OS) and progression-free survival (PFS) in some cancer types, supporting the results of previous publications [102].

Another study, the AcSé trial, showed 50% ORR in the subgroup of *SMARCA4*-deficient rhabdoid tumors, higher than in other rare sarcomas included. One-year PFS and OS were also remarkable (83% and 62.5%, respectively) [103].

Table 2 provides a summary of current phase I and II trials with immunotherapy in patients with rhabdoid tumors harboring SWI/SNF alterations.

#### 4.2.2. Adoptive Cell Therapy

Adoptive cell therapy (ACT) encompasses a range of strategies based on isolation, modification, and infusion of T lymphocytes to overcome tumor evasion and redirect the immune response. The most common modalities of ACT are engineered T cell receptor (TCR) therapy, chimeric antigen receptor (CAR) T cell therapy, and tumor-infiltrating lymphocyte (TIL) therapy [104].

As epigenetic disruption leads to the accumulation of errors in replication, with subsequent mutations, the production of neoantigens increases. These aberrant antigens are derived from tumor cells and can act as potent activators of CD8+ T cell responses [105]. Immunogenicity in rhabdoid tumors with defective SWI/SNFc may derive from the antigens themselves, although the tumor mutational burden (TMB) is another factor to justify T cell infiltration. Despite the hypothetical increased antigen presentation, not all solid tumors with a higher production of self-antigens express high TMB, as can be observed in certain sarcomas, such as synovial and desmoplastic round cell tumors [106].

ACT could be proposed as a new therapeutic option in SWI/SNFc-altered rhabdoid tumors due to its ability to recognize specific antigens. The main feature that makes these solid tumors candidates for ACT is the predominance of monogenic mutations in a high percentage of cases (e.g., *SMARCB1* in ES, MRT or RMC, *SMARCA4* in SCCOHT). In addition, natural T lymphocyte infiltration can be exploited to potentially use ICI with ACT to enhance the cytotoxic response.

TCR therapy is dependent on major histocompatibility complex (MHC) presentation, whereas CAR T and TIL strategies are not. The former requires the isolation of T cells that recognize specific tumor antigens and subsequent transduction of the receptor α and β chains by lentiviral or retroviral vectors. Although this treatment is highly specific, it is not generalizable due to its dependence on the MHC.

CAR T therapy involves the generation of a chimeric molecule combining two fragments: one from an antibody (with recognition capability) and one from the TCR. This avoids the MHC-mediated recognition step, making it applicable to any patient. Currently, CAR T therapies are not approved for the treatment of solid tumors. A major concern is the high rate of toxicity (on-target/off-tumor and cytokine release syndrome) due to the ubiquitous nature of the targeted antigens, which limits their use.

Clinical data on TCR and CAR T strategies in tumors with rhabdoid features are limited to a few phase I and II trials including soft tissue sarcomas, with overall poor results [107,108,109,110,111]. New York esophageal squamous cell carcinoma (NY-ESO)-1 and melanoma-associated antigen (MAGE)-A4 have been selected for engineered TCR (restricted by HLA). To date, no clinical trials have been conducted targeting SWI/SNF antigens.

TIL therapy consists of isolating tumor-specific T lymphocytes within the tumor that recognize tumor-specific antigens. TILs are artificially expanded and later administered following a protocol of lymphodepleting chemotherapy. This strategy has the advantage of using more TCR subpopulations than CAR T and TCR therapies, which target different specific antigens. A number of studies using TILs have been conducted in solid tumors, mainly melanoma [112,113,114,115]. TILs therapy may represent a novel approach for rhabdoid tumors, as TILs could recognize specific aberrations in SWI/SNFc, such as complete loss of *SMARCA4* or *SMARCB1* or other neoantigens created by epigenetic changes occurring within the tumor. Moreover, high lymphocytic infiltration allows for the possibility of expansion if the tumor is amenable to surgery or fresh biopsy.

A multicenter phase II study of TILs treatment in advanced tumors with monogenic SWI/SNFc alterations (the TILTS study) will evaluate the clinical activity of autologous TILs as a single therapy in patients affected by unresectable or advanced rhabdoid tumors (EU CT number 2023-504632-17-00).

## 5. Conclusions

SWI/SNF alterations are present in up to 25% of solid tumors. Their role in oncoge-nesis has attracted preclinical and clinical research, as several cellular processes depend on the correct function of SWI/SNF genes. However, the huge diversity of proteins involved in SWI/SNF makes this novel therapeutic approach challenging.

Rhabdoid tumors are closely associated with SWI/SNF alterations. The lack of effective therapies in the metastatic setting is an unmet clinical need for these patients. As a result, many phase I and II clinical trials have tested drugs targeting a variety of pathways, including SWI/SNF itself, PCR through the EZH2 subunit, or cell cycle kinases. Despite these efforts, only tazemetostat has been approved for the treatment of metastatic or locally advanced epithelioid sarcoma that is ineligible for complete resection. The other therapeutic strategies have not been successful, but a large number of ongoing trials with novel monotherapy or combinatorial approaches will expand our knowledge in this setting.

T cell infiltration and the presence of an immunogenic microenvironment have been described in SWI/SNF-altered rhabdoid tumors. For this reason, immunotherapy has emerged as a promising treatment for these tumors. Indeed, some responses with prolonged survival have been reported. Currently, some early clinical trials are recruiting using anti-PD(L)-1-based strategies. On the other hand, ACT represents a new treatment option for patients with solid tumors harboring defective SWI/SNF subunits. Therefore, further research on ACT (especially TILs therapy) in patients with rhabdoid tumors is warranted.

## Figures and Tables

**Figure 1 ijms-24-11143-f001:**
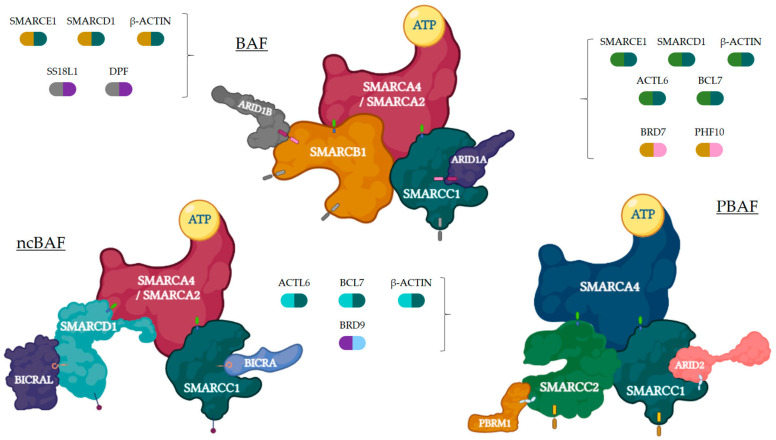
Three SWI/SNFc complexes are represented with their main subunits. Each complex may contain a different core (mustard and dark teal for BAF, turquoise and dark teal for ncBAF, green and dark teal for PBAF) and specific (grey and purple for BAF, sky blue and purple for ncBAF, mustard and pink for PBAF) subunits, apart from the ones presented in the figure, which are represented by bicolor capsules. Figure created with BioRender.com.

**Figure 2 ijms-24-11143-f002:**
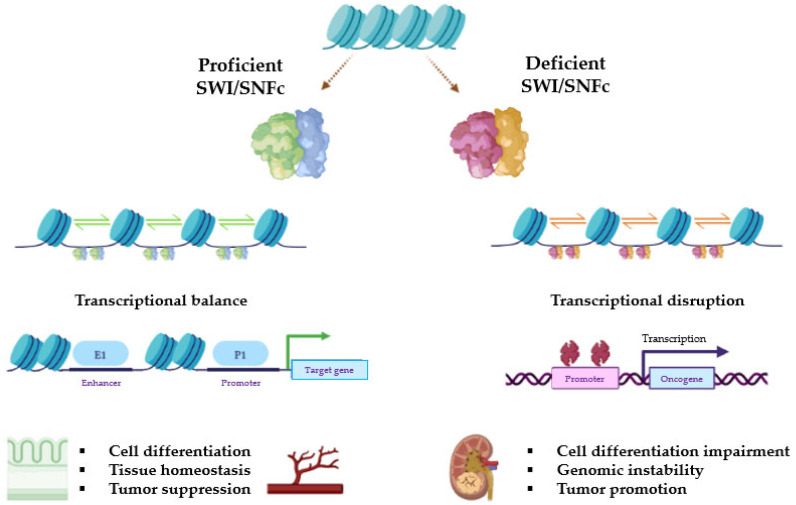
Representative diagram of the transcriptional consequences of whether the SWI/SNFc works properly or not due to deleterious mutations in key subunits. Relative positions among nucleosomes may be pathogenically altered when ATP hydrolysis or histone binding is disrupted. As a result, the accessibility of enhancers provokes different effects on gene transcription. Figure created with BioRender.com.

**Figure 3 ijms-24-11143-f003:**
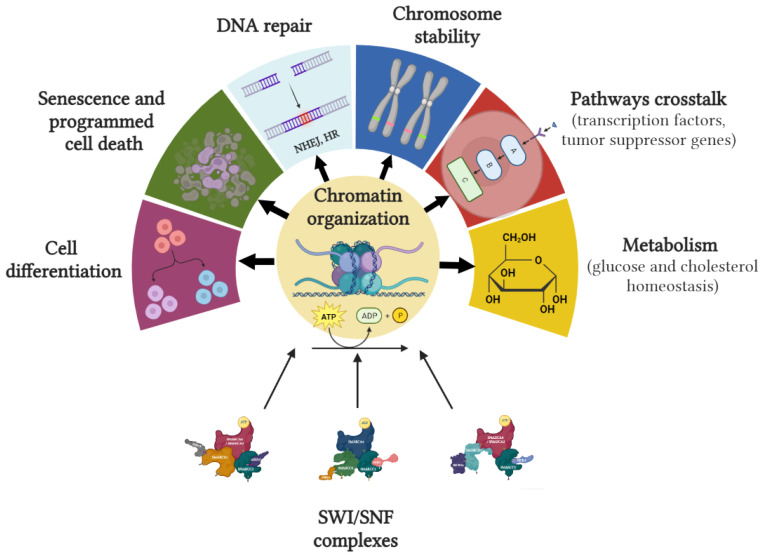
In mammals, SWI/SNFc regulates critical cellular processes, including cell cycle progression, programmed cell death, cell differentiation and development, genomic stability, DNA repair, and metabolism. Figure created with BioRender.com.

**Table 2 ijms-24-11143-t002:** Clinical trials testing ICI, alone or combined with other agents, in rhabdoid and other tumors characterized by specific SWI/SNF alterations.

Author/Year	NCT	Study Design	N	Tumor	Drug	Endpoints/Results and Grade 3–5 AEs
Blay et al. (2019) [103]	03012620	Phase II	21	Rare sarcomas, including rhabdoid and *SMARCA4*-deficient sarcomas	Pembrolizumab	ORR 15%; 1-year PFS 50% (*SMARCA4*-MRT) None reported G3-4 AEs
Ongoing	05286801	Phase I/II	86	Children (1–18 years) with R/R *SMARCB1* or *SMARCA4*-deficient tumors	Tiragolumab and Atezolizumab	Safety (AEs); PK; Efficacy (ORR, PFS, OS, DOR)
Ongoing	05407441	Phase I/II	49	Children and young adults (<24 years) with INI1-neg/*SMARCA4*-deficient tumors	Tazemetostat + Nivolumab/Ipilimumab	Safety (AEs, MTD, RP2D) Efficacy (ORR, OS, PFS)
Ongoing	04416568	Phase II	45	Children, adolescents, and adults with R/R INI1-negative cancers	Nivolumab + Ipilimumab	Efficacy (ORR, PFS, OS, DCR) Safety (AEs)
Ongoing	04284202	Phase II	30	Adults with NSCLC *ARID1*-mutant	Toripalimab + Dasatinib	Efficacy (PFS, OS)
Ongoing	04957615	Phase II	30	Metastatic or unresectable solid tumors with *ARID1A* mutation	Nivolumab	Efficacy (ORR, OS, PFS)
Ongoing	04953104	Phase II	30	Metastatic urothelial cancer with *ARID1A* mutation	Nivolumab	Efficacy (ORR, OS, PFS)

Abbreviations: AEs: Adverse events; DCR: disease control rate; DOR: duration of response; MTD: maximum tolerated dose; ORR: overall response rate; OS: overall survival; PFS: progression-free survival; PK: pharmacokinetics; RP2D: recommended phase 2 dose; R/R: recurrent or relapsed.

## Data Availability

Data sharing not applicable. No new data were created or analyzed in this study. Data sharing is not applicable to this article.

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
