# Peer review of "SWI/SNF Complex Alterations in Tumors with Rhabdoid Features: Novel Therapeutic Approaches and Opportunities for Adoptive Cell Therapy"

_ijms, 2023, doi:10.3390/ijms241311143_

Round 1

Reviewer 1 Report

Overall, the review summarizes a wide swath of the data surrounding the SWI/SNF complex and cancer from a broad array of scientific disciplines – from structural biology to emerging cancer therapies. However, because of the complexity of the topic and the length of the article, the product is somewhat jumbled as it fails to make important connections between topics. This limits the article’s appeal for most audiences.

I would recommend a rewrite to reorganize and refocus the article. Specifically, as the focus is SWI/SNF in cancer, I would recommend starting there and then move to delineating how the SWI/SNF complex is targeted by the therapeutics in Table 1. As the review stands now, these connections are not effectively made. Alternatively, if they cannot be made based on current evidence (e.g. if they are correlative), this also needs to be clearly delineated. A different organization would start with a focus on SMARCB1 / rhaboid tumors (which have the strongest links to SWI/SNF mutations) can be made and then parallels to other forms of cancer can be established. Without this structural rewrite and the connections clearly drawn, the review misses the mark.

Specific issues:

Although the authors cover the fact that the SWI/SNF complex does not bind DNA, they do not mention the Bromodomains in SMARCA2 and SMARCA4 which target these critical subunits to modified histones.

The premise of lines 81-86 is not well explained or established. Is this point necessary for the review?  

The evidence for SWI/SNF being targeted in CART and TIL therapies appears to be lacking. The relative roles of neoantigens vs changes in gene expression downstream of mutations in the SWI/SNF complex are linked to CART and / or TIL therapy needs to be delineated. The references cited do not support a link.

There are numerous typographical or spelling issues (e.g. Maduration in Figure 2) which will have to be addressed in future edits.

Author Response

Point 1: I would recommend a rewrite to reorganize and refocus the article. Specifically, as the focus is SWI/SNF in cancer, I would recommend starting there and then move to delineating how the SWI/SNF complex is targeted by the therapeutics in Table 1. As the review stands now, these connections are not effectively made. Alternatively, if they cannot be made based on current evidence (e.g. if they are correlative), this also needs to be clearly delineated. A different organization would start with a focus on SMARCB1 / rhaboid tumors (which have the strongest links to SWI/SNF mutations) can be made and then parallels to other forms of cancer can be established. Without this structural rewrite and the connections clearly drawn, the review misses the mark.

Response 1: Thank you for your comments. Firstly, we would like to clarify that this review is about SWI/SNF in cancer, but specifically in rhabdoid tumors, with lacking evidence supporting this molecular target for new therapeutic options. That is the reason way we have not written a full revision of SWI/SNF in solid tumors. Secondly, we do not understand the proposal of article’s reestructuration. The review explains what is SWI/SNF complex, what is its role in cancer, which are the rhabdoid tumors with specific alterations and, finally, current approaches to therapeutically target these aberrations. In our opinion, it is written in a neatly and progressive way. Besides, reorganization of paragraphs and information would conflict with other reviewer’s opinion. Based on your comments, we have added information on some SWI/SNF subunits –such as SMARCB1 or SMARCA4-, and linked them with other solid tumors. Different ways of targeting are explained at point 4 -“Therapeutic approaches”-.

Point 2: Although the authors cover the fact that the SWI/SNF complex does not bind DNA, they do not mention the Bromodomains in SMARCA2 and SMARCA4 which target these critical subunits to modified histones.

Response 2: Thank you. We have reviewed this issue and added information accordingly.

Point 3: The premise of lines 81-86 is not well explained or established. Is this point necessary for the review? 

Response 3: Thank you. We have rewritten the paragraph, linking ideas with the next one. The information is related.

Point 4: The evidence for SWI/SNF being targeted in CART and TIL therapies appears to be lacking. The relative roles of neoantigens vs changes in gene expression downstream of mutations in the SWI/SNF complex are linked to CART and / or TIL therapy needs to be delineated. The references cited do not support a link.

Response 4: Thank you. We have extended the rationale and explained better why adoptive cell therapies may be used harnessing SWI/SNF alterations.

Point 5: There are numerous typographical or spelling issues (e.g. Maduration in Figure 2) which will have to be addressed in future edits.

Response 5: Thank you. We have reviewed the entire article and doubled checked the mistakes with a native physician.

Reviewer 2 Report

This is a very well written review on SWI/ SNF alterations. It details the structure and fuction of the complex in physiologial and pathological contexts, detailing the most common alterations and the rate in which they occur, pointing out the current therapeutioc strategies in development that touch upon this. I learned a lot while reading the manuscript and I'm sure it will be a welcome addition to the literature.

Overall its easy to read and comprehend, but there are few minor issues here and there, such as:

"Even with such a variable composition, the structure of the complexes are broadly conserved" should read as "Even with such a variable composition, the structure of the complexes is broadly conserved"

"there is a strong rationale for accelerate research and find therapeutic 195 approaches targeting SWI/SNF pathogenic aberrations." shoud read as "there is a strong rationale for accelerating research and finding therapeutic approaches targeting SWI/SNF pathogenic aberrations."

Author Response

Point 1: Overall its easy to read and comprehend, but there are few minor issues here and there, such as:

"Even with such a variable composition, the structure of the complexes are broadly conserved" should read as "Even with such a variable composition, the structure of the complexes is broadly conserved"

"there is a strong rationale for accelerate research and find therapeutic 195 approaches targeting SWI/SNF pathogenic aberrations." shoud read as "there is a strong rationale for accelerating research and finding therapeutic approaches targeting SWI/SNF pathogenic aberrations."

Response 1: Thank you for your comments. We have reviewed the entire text with a native physician. Some phrases and paragraphs have been modified accordingly.

"Even with such a variable composition, the structure of the complexes are broadly conserved" has been changed to "Despite this variable composition, the structure of the complexes is largely conserved ".

"There is a strong rationale for accelerate research and find therapeutic 195 approaches targeting SWI/SNF pathogenic aberrations" has been changed to "there is a strong rationale for accelerating research and finding therapeutic approaches that target SWI/SNF aberrations".

Reviewer 3 Report

Juan et al. comprehensively reviewed the alterations of the SWI/SNF complex in rhabdomyosarcoma and its potential applications in tumor therapy. This review is highly significant as the study of the SWI/SNF complex in tumors is an emerging field. The occurrence rate of gene mutations encoding SWI/SNF is remarkably high, with approximately 25% of tumor patients carrying mutations. Therefore, it serves as an important immunological checkpoint and therapeutic target in current cancer research. The logical coherence of this review is strong, and the schematic diagrams provided are easily comprehensible. Additionally, the review incorporates numerous recent references, which greatly aids readers in quickly understanding the progress made in the field of SWI/SNF. Furthermore, the authors need to address the following issues to enhance the quality of this article:

1. Summarize the mutations and expression changes of SWI/SNF-related genes in rhabdomyosarcoma.

2. Investigate whether SWI/SNF-related genes can serve as biomarkers for rhabdomyosarcoma.

3. Explore the mechanisms by which SWI/SNF loss contributes to the development of rhabdomyosarcoma.

4. Correct minor grammatical errors and thoroughly proofread the entire manuscript.

Correct minor grammatical errors and thoroughly proofread the entire manuscript.

Author Response

Point 1: Furthermore, the authors need to address the following issues to enhance the quality of this article:

Summarize the mutations and expression changes of SWI/SNF-related genes in rhabdomyosarcoma.

Response 1: Thank you for your suggestions. We have reviewed bibliography about rhabdomyosarcoma and related SWI/SNF alterations. However, there is scarce mention about SMARCA4 and BAF47 protein (SMARCB1 gene) alterations -cites attached-. The latest appears in an article published in 1999, with low impact on clinical practice due to findings in cell lines. We have summarised the main findings. However, this is not a specific tumor characterized by SWI/SNF alterations.

Point 2: Investigate whether SWI/SNF-related genes can serve as biomarkers for rhabdomyosarcoma.

Response 2: Thank you. We have written a brief comment focused on synthetic lethality. However, we have not found any clinical study describing SWI/SNF alterations as biomarkers of response or prognosis in rhabdomyosarcoma.

Point 3: Explore the mechanisms by which SWI/SNF loss contributes to the development of rhabdomyosarcoma.

Response 3: Thank you. Mechanisms are broadly described in a whole section of the article., although not especifically refered to rhabdomyosarcoma. In the new version we have added a brief comment on SMARCA4 contribution.

Point 4: Correct minor grammatical errors and thoroughly proofread the entire manuscript.

Response 4: Thank you. We have revised the entire article with a native physician.
